# Tetrahydrofurofuranoid Lignans, Eudesmin, Fargesin, Epimagnolin A, Magnolin, and Yangambin Inhibit UDP-Glucuronosyltransferase 1A1 and 1A3 Activities in Human Liver Microsomes

**DOI:** 10.3390/pharmaceutics13020187

**Published:** 2021-02-01

**Authors:** Ria Park, Eun Jeong Park, Yong-Yeon Cho, Joo Young Lee, Han Chang Kang, Im-Sook Song, Hye Suk Lee

**Affiliations:** 1College of Pharmacy, The Catholic University of Korea, Bucheon 14662, Korea; hyacinthy7@catholic.ac.kr (R.P.); enddl0818@catholic.ac.kr (E.J.P.); yongyeon@catholic.ac.kr (Y.-Y.C.); joolee@catholic.ac.kr (J.Y.L.); hckang@catholic.ac.kr (H.C.K.); 2College of Pharmacy and Research Institute of Pharmaceutical Sciences, Kyungpook National University, Daegu 41566, Korea; isssong@knu.ac.kr

**Keywords:** eudesmin, fargesin, epimagnolin A, magnolin, yangambin, human liver microsomes, UDP-glucuronosyltransferase

## Abstract

Eudesmin, fargesin, epimagnolin A, magnolin, and yangambin are tetrahydrofurofuranoid lignans with various pharmacological activities found in Magnoliae Flos. The inhibition potencies of eudesmin, fargesin, epimagnolin A, magnolin, and yangambin on six major human uridine 5′-diphospho-glucuronosyltransferase (UGT) activities in human liver microsomes were evaluated using liquid chromatography–tandem mass spectrometry and cocktail substrates. Eudesmin, fargesin, epimagnolin A, magnolin, and yangambin inhibited UGT1A1 and UGT1A3 activities, but showed negligible inhibition of UGT1A4, UGT16, UGT1A9, and UGT2B7 activities at 200 μM in pooled human liver microsomes. Moreover, eudesmin, fargesin, epimagnolin A, magnolin, and yangambin noncompetitively inhibited UGT1A1-catalyzed SN38 glucuronidation with *K_i_* values of 25.7, 25.3, 3.6, 26.0, and 17.1 μM, respectively, based on kinetic analysis of UGT1A1 inhibition in pooled human liver microsomes. Conversely, the aforementioned tetrahydrofurofuranoid lignans competitively inhibited UGT1A3-catalyzed chenodeoxycholic acid 24-acyl-glucuronidation with 39.8, 24.3, 15.1, 37.6, and 66.8 μM, respectively in pooled human liver microsomes. These in vitro results suggest the necessity of evaluating whether the five tetrahydrofurofuranoid lignans can cause drug–drug interactions with UGT1A1 and UGT1A3 substrates in vivo.

## 1. Introduction

Eudesmin, fargesin, epimagnolin A, magnolin, and yangambin (Figure 1) are the pharmacologically active tetrahydrofurofuranoid lignans found in *Magnolia denudata* Desrousseaux, *M. fargesii* Cheng (family: Magnoliaceae), and *Zanthoxylum armatum* DC (family: Rutaceae) [1,2,3,4,5]. These lignans possess anti-inflammatory activity through the inhibitory effects on 5-lipoxygenase and nitric oxide synthase [5,6,7,8,9] and antitumor activity [10,11,12]. In addition to the anti-inflammatory and antitumor activity of tetrahydrofurofuranoid lignans, eudesmin showed bactericidal effect on Helicobacter pylori reference strain 26,695 and suppressed inflammatory and immune responses in Helicobacter pylori-infected mice [13]. Fargesin exhibits additional biological activities, including cardioprotective effect in rats with ischemia/reperfusion injury [14] and lipid lowering and hypoglycemic effects in high-fat diet-induced obese mice [15], and antihypertensive effects in 2K1C hypertensive rats [16]. Magnolin also ameliorates contrast-induced nephropathy in rats via anti-oxidation and anti-apoptosis [17].

The ethanol extract of the dried flower buds of *M. fargesii* (encoded as NDC-052 contained eudesmin, fargesin, epimagnolin A, magnolin, and yangambin as 4.1, 3.4, 11.9, 21.5, and 9.1%, respectively, as determined by the LC-APCI-MS/MS method [18]) has been developed as an effective alternative or complement to standard asthma therapy based on their biological activities [19]. Consequently, taking NDC-052 (600 mg/day for 8 weeks per oral) in adult asthmatic patients was safe and tolerated [19]. The add-on therapy of NDC-052 (600 mg/day for 8 weeks per oral) with inhaled corticosteroids in asthmatic patients had a beneficial effect on asthma control [19]. The pharmacokinetics of herbal constituents in rats have been investigated in addition to their biological activities. Eudesmin, yangambin, epimagnolin A, fargesin, and magnolin were identified in rat plasma following oral administration (5.5–22 mg/kg) of NDC-052 to rats [18]. The area under the plasma concentration curve (AUC) and maximum plasma concentration (C_max_) values of eudesmin, yangambin, and epimagnolin linearly increased with dose increase [18]. However, AUC and C_max_ values of fargesin and magnolin were not increased with dose proportionality, suggesting the nonlinear pharmacokinetic properties of fargesin and magnolin [18]. Additionally, the AUC and C_max_ values of magnolin in rats were 7568 ± 1085 ng·h/mL and 2493 ± 513 ng/mL, respectively, when administered NDC-052 (22.2 mg/kg containing 4 mg/kg of magnolin) [20] and had similar results with the previous report [18]. However, the AUC and C_max_ values of magnolin in rats were 3630 ± 581 ng·h/mL and 1340 ± 113 ng/mL, respectively, when administered magnolin alone (4 mg/kg) [21]. These results suggested the possibility of drug interactions among the constituents of herbal drugs.

The herb-drug interactions among the herbal drugs (e.g., *Hypericum perforatum*, *Ginkgo biloba*, *Camellia sinensis*, *Glycyrrhiza glabra*, *Allium sativum*, *Coptis chinensis*, and *Silybum marianum*) and their constituents via the inhibition or induction of major drug-metabolizing enzymes, cytochrome P450 (CYP) and uridine 5′-diphospho (UDP)-glucuronosyltransferase (UGT) have been reported [22,23,24,25,26,27]. Thus, evaluating the in vitro inhibitory effects of the active components of herbal drugs on human CYP and UGT activities is necessary for the assessment of herb-drug interactions.

Aschantin, a bioactive tetrahydrofurofuranoid lignan found in *M. biondii* and *Hernandia nymphaeifolia*, exhibited the time-dependent inhibition of CYP2C8 (*K_i_*: 10.2 µM and *k*_inact_: 0.056 min^−1^), CYP2C9 (*K_i_*: 3.7 µM and *k*_inact_: 0.044 min^−1^), CYP2C19 (*K_i_*: 5.8 µM and *k*_inact_: 0.048 min^−1^), and CYP3A4 (*K_i_*: 12.6 µM and *k*_inact_: 0.062 min^−1^) in human liver microsomes [28]. The inactivation efficiency (*k*_inact_/*K*_i_) of aschantin on CYP2C8, CYP2C9, CYP2C19, and CYP3A4 activities was calculated as 5.49, 11.9, 8.28, 4.92 mL/μmol/min, respectively, which were in the range of the inactivation efficiency of clarithromycin (2.8 mL/μmol/min) and verapamil (11.2 mL/μmol/min) on CYP3A4, representative examples of clinical CYP3A4-mediated drug interactions [29]. Aschantin weakly inhibited the catalytic activities of UGT1A1 (IC_50_, 131.7 µM), UGT1A6 (IC_50_, 144.1 µM), and UGT1A9 (IC_50_, 71.0 µM) in human liver microsomes [28].

The inhibitory effects of yangambin, epimagnolin A, eudesmin, fargesin, and magnolin on CYP activities in human liver microsomes were previously investigated [30]. Fargesin inhibited competitively CYP2C9-catalyzed diclofenac 4′-hydroxylation (*K*_i_, 16.3 μM) [30]. It also exhibited the time-dependent inhibition of CYP2C8 (*K_i_*: 10.7 µM and *k*_inact_: 0.082 min^−1^), CYP2C19 (*K_i_*: 3.7 µM and *k*_inact_: 0.102 min^−1^), and CYP3A4 (*K_i_*: 23.0 µM and *k*_inact_: 0.050 min^−1^) in human liver microsomes. However, fargesin negligibly inhibited the catalytic activities of CYP1A2, CYP2A6, CYP2B6, and CYP2D6. In addition, eudesmin, epimagnolin A, magnolin, and yangambin slightly inhibited eight major CYP activities in human liver microsomes, indicating that they may not cause CYP-mediated drug interactions [30]. However, no reports exist on in vitro and in vivo inhibitory effects of eudesmin, fargesin, epimagnolin A, magnolin, and yangambin on human UGT enzymes.

Therefore, this study aimed to investigate the in vitro inhibitory potentials and inhibition kinetics of eudesmin, fargesin, epimagnolin A, magnolin, and yangambin, that are major tetrahydrofurofuranoid lignans contained in NDC-052 and also found in rat plasma samples [18], on UGT1A1, UGT1A3, UGT1A4, UGT1A6, UGT1A9, and UGT2B7 activities using pooled human liver microsomes to provide the underlying drug interaction potentials of eudesmin, fargesin, epimagnolin A, magnolin, and yangambin.

## 2. Materials and Methods

### 2.1. Materials and Reagents

Fargesin (purity, 99.2%) and magnolin (purity, 98.9%) were obtained from PhytoLab GmbH & Co. (Vestenbergsgreuth, Germany). Eudesmin (purity, 99%) was purchased from Selleck Chemicals LLC. (Houston, TX, USA). Epimagnolin A (purity, 98%) and yangambin (purity, 98%) were obtained from ChemFaces Biochemical Co. Ltd. (Wuhan, China). Ultrapooled human liver microsomes (catalog number 452161) were obtained from Corning Life Sciences (Woburn, MA, USA). Alamethicin, UDP-glucuronic acid (UDPGA), Trizma base, *N*-acetylserotonin, chenodeoxycholic acid, ketoconazole, naloxone, naloxone 3-β-D-glucuronide, mycophenolic acid, and trifluoperazine dihydrochloride were provided by Sigma-Aldrich (St. Louis, MO, USA). Chenodeoxycholic acid 24-acyl-β-glucuronide, SN-38 glucuronide, and mycophenolic acid β-D-glucuronide were provided by Toronto Research Chemicals (Toronto, ON, Canada). SN-38 was provided by Santa Cruz Biotechnology (Dallas, TX, USA). Water, methanol, and acetonitrile of high-performance liquid chromatography grade were provided by Fischer Scientific (Fair Lawn, NJ, USA).

### 2.2. Inhibitory Potentials of Eudesmin, Fargesin, Epimagnolin A, Magnolin, and Yangambin on Six Major UGT Activities in Ultrapooled Human Liver Microsomes

The inhibitory potentials of eudesmin, fargesin, epimagnolin A, magnolin, and yangambin on six UGT activities were measured as described previously [31]. Briefly, reaction mixture (100 µL) containing 50 mM Tris buffer (pH 7.4), 10 mM MgCl_2_, 25 µg/mL alamethicin, 5 mM UDPGA, ultrapooled human liver microsomes (0.2 mg/mL), various concentrations of eudesmin, fargesin, epimagnolin A, magnolin, or yangambin in acetonitrile (final concentrations of 0.1, 1, 10, 20, 50, 100, and 200 µM each, acetonitrile less than 1% (*v*/*v*), and the UGT probe cocktails of set A (0.5 µM SN-38 for UGT1A1, 2 µM chenodeoxycholic acid for UGT1A3, and 0.5 µM trifluoperazine for UGT1A4) or set B (1 µM *N*-acetylserotonin for UGT1A6, 0.2 µM mycophenolic acid for UGT1A9, and 1 µM naloxone for UGT2B7) were incubated at 37 °C for 60 min. Fifty microliters of meloxicam and propofol glucuronide (internal standards, ISs) in acetonitrile was added to the reaction tube, and the mixtures were centrifuged at 13,000× *g* for 4 min at 4 °C. Fifty microliters of the supernatants from set A and set B reaction tubes were mixed, and 3 µL-aliquot of the mixture was analyzed by LC-MS/MS.

### 2.3. Kinetic Analysis

The incubation mixtures (100 µL) including 50 mM Tris buffer (pH 7.4), human liver microsomes (0.1 mg/mL), various concentrations of SN-38 (0.2–2 µM) for UGT1A1 or chenodeoxycholic acid (1–10 µM) for UGT1A3, 5 mM UDPGA, 25 µg/mL alamethicin, 10 mM MgCl_2_, and various concentrations of eudesmin (2–40 µM), fargesin (2–30 µM), epimagnolin A (0.2–5 µM for UGT1A1; 2–30 µM for UGT1A3), magnolin (2–40 µM), or yangambin (2–40 µM for UGT1A1; 5–80 µM for UGT1A3) were incubated for 30 min to determine the *K_i_* values and inhibition mode for inhibition of eudesmin, fargesin, epimagnolin A, magnolin, and yangambin on UGT1A1 and UGT1A3 activities. The inhibitory effect of NDC-052 on UGT1A1 and UGT1A3 activities was also measured by incubating the incubation mixtures (100 µL) including 50 mM Tris buffer (pH 7.4), human liver microsomes (0.1 mg/mL), SN-38 (0.5 µM) for UGT1A1 or chenodeoxycholic acid (2 µM) for UGT1A3, 5 mM UDPGA, 25 µg/mL alamethicin, 10 mM MgCl_2_, and various concentrations of ND-052 (0.1–100 µg/mL) for 30 min. The reactions were stopped by adding 100 μL of 500 ng/mL meloxicam (IS for SN-38 glucuronide) or propofol glucuronide (IS for chenodeoxycholic acid 24-acyl-β-glucuronide) in ice-cold acetonitrile. The incubation mixtures were then centrifuged at 13,000 × *g* for 5 min, and 30 μL of the supernatant was diluted with 70 μL of water. Aliquots (3 μL) were analyzed by LC-MS/MS.

### 2.4. LC-MS/MS Analysis

UGT metabolites were simultaneously quantified using Agilent 6495 triple quadrupole mass spectrometer equipped with Agilent 1290 Infinity UPLC (Agilent Technologies, Wilmington, DE, USA) as described previously [31]. Analytes were eluted from an Atlantis dC18 column (3 µm, 2.1 mm internal diameter × 100 mm, Waters Co., Milford, MA, USA) by the gradient elution of mobile phase A (5% acetonitrile in 0.1% formic acid) and mobile phase B (95% acetonitrile in 0.1% formic acid) at flow rate of 0.3 mL/min. The electrospray ionization source settings in both positive and negative ion modes were drying gas flow, 14 L/min; nebulizer pressure, 40 psi; sheath gas flow, 11 L/min; drying gas temperature, 200 °C; sheath gas temperature, 380 °C; nozzle voltage, 500 V; capillary voltage, 4500 V; and fragmentor voltage, 380 V. Selected reaction monitoring (SRM) transitions and collision energy in the positive ion mode were *m*/*z* 503.9→309.9 and 32 eV for naloxone 3-β-D-glucuronide, *m*/*z* 394.9→219.0 and 10 eV for *N*-acetylserotonin β-D-glucuronide, *m*/*z* 583.9→407.9 and 26 eV for trifluoperazine glucuronide, *m*/*z* 568.9→392.9 and 30 eV for SN-38 glucuronide, and *m*/*z* 351.9→115.0 and 20 eV for meloxicam (IS). SRM transitions and collision energy in the negative ion mode: *m*/*z* 567.1→391.2 and 34 eV for chenodeoxycholic acid 24-acyl-β-glucuronide, *m*/*z* 495.0→319.0 and 20 eV for mycophenolic acid glucuronide, and *m*/*z* 353.0→177.0 and 28 eV for propofol glucuronide (IS). The linear concentrations for trifluoperazine glucuronide were 4–1200 pmol, but those of five other UGT metabolites were 1–300 pmol. The relative error and coefficient of variation values for quality control samples of six metabolites were −2.5–3.0% and 5.9–9.8%.

### 2.5. Data Analysis

IC_50_ values (i.e., the inhibitor concentrations required for 50% inhibition of the control activity) of eudesmin, fargesin, epimagnolin A, magnolin, and yangambin on UGT activities were estimated using SigmaPlot (version 12.0; Systat Software, San Jose, CA, USA). The apparent kinetic inhibition constant (*K_i_*) values of eudesmin, fargesin, epimagnolin A, magnolin, and yangambin and the mode of inhibition were calculated from Dixon plot transformation [31].

## 3. Results

### 3.1. Inhibitory Effect of Eudesmin on Human Uridine 5′-diphospho-glucuronosyltransferase (UGT) Isoforms

The inhibitory effects of five tetrahydrofurofuranoid lignans on six major human UGT isoforms were measured in ultrapooled human liver microsomes and the IC_50_ values calculated from the concentration dependent inhibition curves of UGT activities are summarized in Table 1.

The UGT1A1 and UGT1A3 activities were inhibited by the presence of eudesmin (0.1–200 µM) in a concentration dependent manner with IC_50_ values of 24.3 and 26.6 µM, respectively, when measured by the formation of SN-38 glucuronide and chenodeoxycholic acid 24-acyl-glucuronide. The UGT1A6 and UGT1A9 activities were slightly inhibited by the presence of eudesmin with IC_50_ values of 195.6 and 173.2 µM, respectively, when measured by the formation of *N*-acetylserotonin glucuronide and mycophenolic acid glucuronide. However, UGT1A4 and UGT2B7 activities were not inhibited by eudesmin up to 200 µM tested (Figure 2, Table 1).

### 3.2. Inhibitory Effect of Fargesin on Human UGT Isoforms

The UGT1A1 and UGT1A3 activities were inhibited by the presence of fargesin (0.1–200 µM) in a concentration dependent manner with IC_50_ values of 24.5 and 21.5 µM, respectively, when measured by the formation of SN-38 glucuronide and chenodeoxycholic acid 24-acyl-glucuronide, respectively. The metabolic activities of UGT1A4, UGT1A6, UGT1A9, and UGT2B7 were slightly inhibited by the presence of fargesin with IC_50_ values of 182.7, 193.9, 110.9, and 94.7 µM, respectively (Figure 3, Table 1).

### 3.3. Inhibitory Effect of Epimagnolin A on Human UGT Isoforms

The UGT1A1 and UGT1A3 activities were inhibited by the presence of epimagnolin A (0.1–200 µM) in a concentration dependent manner with IC_50_ values of 7.5 and 26.6 µM, respectively, when measured by the formation of SN-38 glucuronide and chenodeoxycholic acid 24-acyl-glucuronide. The metabolic activities of UGT1A4, UGT1A6, UGT1A9, and UGT2B7 were negligibly inhibited by epimagnolin A up to 200 µM tested (Figure 4, Table 1).

### 3.4. Inhibitory Effect of Magnolin on Human UGT Isoforms

The UGT1A1 and UGT1A3 activities were inhibited by the presence of magnolin (0.1–200 µM) in a concentration dependent manner with IC_50_ values of 21.3 and 22.9 µM, respectively, when measured by the formation of SN-38 glucuronide and chenodeoxycholic acid 24-acyl-glucuronide. The UGT1A9 activity was slightly inhibited by the presence of magnolin with IC_50_ values of 145.7 µM, when measured by the formation of mycophenolic acid glucuronide. However, the UGT1A4, UGT1A6, and UGT2B7 activities were not inhibited by the presence of magnolin up to 200 µM tested (Figure 5, Table 1).

### 3.5. Inhibitory Effect of Yangambin on Human UGT Isoforms

The UGT1A1 and UGT1A3 activities were inhibited by the presence of yangambin (0.1–200 µM) in a concentration dependent manner with IC_50_ values of 29.7 and 56.5 µM, respectively, when measured by the formation of SN-38 glucuronide and chenodeoxycholic acid 24-acyl-glucuronide. The metabolic activities of UGT1A4, UGT1A6, UGT1A9, and UGT2B7 were negligibly inhibited by yangambin up to 200 µM tested (Figure 6, Table 1).

### 3.6. Kinetic Analysis of Eudesmin, Fargesin, Epimagnolin A, Magnolin, and Yangambin on UGT1A1 and UGT1A3 Activities in Human Liver Microsomes

The Dixon plots were used to calculate the *K_i_* values and inhibition modes of eudesmin, fargesin, epimagnolin A, magnolin, and yangambin for the inhibition of catabolic activities of UGT1A1 and UGT1A3 enzymes (Figure 7 and Figure 8, Table 2). In addition, eudesmin, fargesin, epimagnolin A, magnolin, and yangambin noncompetitively inhibited SN-38 glucuronidation by UGT1A1 with *K_i_* values of 25.7, 25.3, 3.6, 26.0, and 17.1 μM, respectively, in pooled human liver microsomes (Figure 7, Table 2). Conversely, eudesmin, fargesin, epimagnolin A, magnolin, and yangambin showed the competitive inhibition of chenodeoxycholic acid 24-acyl-glucuronidation by UGT1A3 with *K_i_* values of 39.8, 24.5, 15.1, 37.6, and 66.8 μM, respectively (Figure 8, Table 2).

Next, we measured the IC_50_ values of NDC-052 that contained eudesmin, fargesin, epimagnolin A, magnolin, and yangambin at a content of 4.1, 3.4, 11.9, 21.5, and 9.1%, respectively [18] on UGT1A1- and UGT1A3-mediated glucuronidation. The UGT1A1 and UGT1A3 activities were inhibited by the presence of NDC-052 (0.1–100 µg/mL) in a concentration dependent manner and the IC_50_ values of NDC-052 for UGT1A1 and UGT1A3 were calculated as 38.1 and 65.0 µg/mL, respectively.

## 4. Discussion

Eudesmin, fargesin, epimagnolin A, magnolin, and yangambin were selected as major tetrahydrofurofuranoid lignans that exist in the ethanol extract of flower bud of *M. fargesii* (NDC-052) and are also present in the rat plasma samples following oral administration [18]. Therefore, the inhibitory potentials (IC_50_ values) of eudesmin, fargesin, epimagnolin A, magnolin, and yangambin on UGT1A1, UGT1A3, UGT1A4, UGT1A6, UGT1A9, and UGT2B7 enzyme activities were evaluated in pooled human liver microsomes in the concentration ranges of 0.1–200 μM of individual tetrahydrofurofuranoid lignan (Figure 2, Figure 3, Figure 4, Figure 5 and Figure 6; Table 1). The concentration range of lignans tested in this study could cover the plasma concentration of lignans in rats after an oral administration of NDC-052 (22 mg/kg) and the expected gastric concentrations of five lignans. For example, the C_max_ values of eudesmin, fargesin, epimagnolin A, magnolin, and yangambin were 2.2 ± 0.41 μM, 0.38 ± 0.18 μM, 3.07 ± 0.72 μM, 5.95 ± 1.12 μM, and 0.52 ± 0.30 μM, respectively, after single oral administration of NDC-052 (22 mg/kg) [18]. Herbal extracts are diluted by gastrointestinal fluid after oral administration by a factor of oral dilution volume (500–2200 mL in humans) [32,33]. Therefore, the expected gastric concentrations of eudesmin, fargesin, epimagnolin A, magnolin, and yangambin could be calculated as 29–127 μM, 25–110 μM, 78–342 μM, 140–618 μM, and 55–244 μM by considering the content of five lignans (eudesmin, fargesin, epimagnolin A, magnolin, and yangambin as 4.1, 3.4, 11.9, 21.5, and 9.1%, respectively) [18] and oral dose of NDC-052 in adult asthmatic patients (600 mg/day for 8 weeks) [19].

Eudesmin, fargesin, epimagnolin A, magnolin, and yangambin reversibly and noncompetitively inhibited UGT1A1-catalyzed SN-38 glucuronidation with *K_i_* values of 25.7, 25.3, 3.6, 26.0, and 17.1 μM, respectively, indicating the possible interactions with UGT1A1 substrates (e.g., bilirubin, irinotecan, trovafloxacin, belinostat, dolutegravir, pterostilbene, and resveratrol) [34,35,36,37,38]. In addition, eudesmin, fargesin, epimagnolin A, magnolin, and yangambin were competitive inhibitors of UGT1A3 with *K_i_* values of 39.8, 24.5, 15.1, 37.6, and 66.8 μM, respectively, indicating that they should be used carefully with UGT1A3 substrates (e.g., chenodeoxycholic acid, fimasartan, losartan, candesartan, and zolarsartan) [39,40,41] to avoid possible drug interactions. However, they showed weak or negligible inhibition toward the other four UGT isoforms such as UGT1A4, UGT1A6, UGT1A9, and UGT2B7 tested in this study (IC_50_ values of about 100 µM or higher). The five tetrahydrofurofuranoid lignans possess a similar chemical structure, which made reversible inhibition of five lignans to UGT1A1 and UGT1A3 but not to UGT1A4, UGT1A6, UGT1A9, and UGT2B7 isozymes. Moreover, these five tetrahydrofurofuranoid lignans were all found in NDC-052 with C_max_ values of 2.2 ± 0.41 μM (eudesmin), 0.38 ± 0.18 μM (fargesin), 3.07 ± 0.72 μM (epimagnolin A), 5.95 ± 1.12 μM (magnolin), and 0.52 ± 0.30 μM (yangambin), respectively, in rats [18]. Therefore, these five lignans may potentiate coordinately the likelihood of UGT1A1- and UGT1A3-mediated drug interaction with co-administration of victim drugs. 

As expected, NDC-052 inhibited UGT1A1- and UGT1A3-mediated glucuronidation in a concentration dependent manner with IC_50_ values of 38.1 and 65.0 µg/mL, respectively. When the IC_50_ values were calculated as the concentration of individual lignans by using the content of five lignans in NDC-052, 38.1 and 65.0 µg/mL of NDC-052 contained 4.0 and 6.9 µM of eudesmin, 3.5 and 6.0 µM of fargesin, 10.9 and 18.6 µM of epimagnolin A, 19.7 and 33.6 µM of magnolin, and 7.8 and 13.2 µM of yangambin, respectively. These concentrations were lower than the IC_50_ values of individual lignans in Table 1, which suggested that the inhibitory effect of NDC-052 on the metabolic activities of UGT1A1 and UGT1A3 may be caused by the coexistence of lignans rather than the inhibition by single lignan. However, we also should note that the total content of five lignans accounts for 50% of NDC-052 and unveiled components such as terpenoids, alkaloids, and flavonoids may serve as inhibitors of UGT enzymes [42].

The efficacy of NDC-052 has been evaluated for the treatment of asthma in guinea pig chronic asthma model at a repeated oral dose (50 mg/kg/day for 8 weeks) as well as in adult asthmatic patients (600 mg/day for 8 weeks) [19]. Although the plasma and gastric concentrations of individual eudesmin, fargesin, epimagnolin A, magnolin, and yangambin in asthmatic patients were not reported, the sum of the plasma and gastric concentrations for these five lignans may reach or exceed *K_i_* values to UGT1A1 and UGT1A3 in the above efficacy model considering the C_max_ of the previous study [18] and the expected gastric concentration of five lignans after repeated oral administration. Therefore, these five lignans may potentiate coordinately the likelihood of UGT1A1- and UGT1A3-mediated drug interaction with co-administration of victim drugs.

In addition, among the five tetrahydrofurofuranoid lignans (e.g., eudesmin, epimagnolin A, fargesin, magnolin, and yangambin), only fargesin inhibited competitively CYP2C9 (*K_i_*, 16.3 μM) and irreversibly inhibited CYP2C8 (*K_i_*: 10.7 µM and *k*_inact_: 0.082 min^−1^), CYP2C19 (*K_i_*: 3.7 µM and *k*_inact_: 0.102 min^−1^), and CYP3A4 (*K_i_*: 23.0 µM and *k*_inact_: 0.050 min^−1^) in human liver microsomes [22]. Consequently, it indicated that CYP2C8, CYP2C9, CYP2C19, and CYP3A4 activities might be inhibited by the presence of fargesin that may cause reversible inhibition on CYP2C9 and the formation of intermediate metabolites of fargesin which may cause irreversible inhibition ofCYP2C8, CYP2C19, and CYP3A4. This could cause CYP2C8, CYP2C9, CYP2C19, and CYP3A4-mediated drug interaction with co-administered victim drugs or co-existing components such as magnolin, epimagnolin A, eudesmin, and yangambin.

In a previous study, the AUC of magnolin was linearly increased with an increase in the intravenous dose (0.5–2 mg/kg) and in the oral dose (1–4 mg/kg) [21]. In addition, magnolin was mainly eliminated by metabolism [21]. Three major metabolites (i.e., *O*-desmethylmagnolin, i.e., M1 and M2, and hydroxymagnolin, M4) were formed by CYP2C8, CYP2C9, CYP2C19, and CYP3A4 in the in vitro studies of magnolin metabolism [43]. However, the AUC of magnolin was not proportionally increased with increasing oral dose of NDC-052 (5.5–22 mg/kg containing 1–4 mg/kg magnolin) and about two-fold higher than that following oral administration of magnolin alone [18,20]. Taken together, fargesin may be involved in the two-fold increase of magnolin AUC by inhibiting the CYP2C8, CYP2C9, CYP2C19, and CYP3A4-mediated magnolin metabolism when orally administered as NDC-052 to the same dose of single magnolin. For the elucidation of the coordinated drug interaction among the similar structure possessing components such as these tetrahydrofurofuranoid lignans in the NDC-052, the CYPs and UGTs-mediated metabolism of these tetrahydrofurofuranoid lignans awaits further investigation. In this regard, this study would provide the molecular mechanism for understanding the drug interactions among the structurally related herbal components and the concomitantly administered victim drugs.

## 5. Conclusions

Eudesmin, fargesin, epimagnolin A, magnolin, and yangambin noncompetitively inhibited UGT1A1-catalyzed SN-38 glucuronidation with *K_i_* values of 25.7, 25.3, 3.6, 26.0, and 17.1 μM, respectively. Conversely, the aforementioned tetrahydrofurofuranoid lignans competitively inhibited UGT1A3-catalyzed chenodeoxycholic acid 24-acyl-glucuronidation with *K_i_* values of 39.8, 24.5, 15.1, 37.6, and 66.8 μM, respectively. However, these five tetrahydrofurofuranoid lignans weakly or negligibly inhibited the glucuronidation activities of UGT1A4, UGT1A6, UGT1A9, and UGT2B7. Consequently, these in vitro results suggest that eudesmin, fargesin, epimagnolin A, magnolin, and yangambin alone and the herbal preparations containing these tetrahydrofurofuranoid lignans should be examined in terms of potential in vivo pharmacokinetic drug interactions caused by inhibition of UGT1A1 and UGT1A3 activities.

## Figures and Tables

**Figure 1 pharmaceutics-13-00187-f001:**
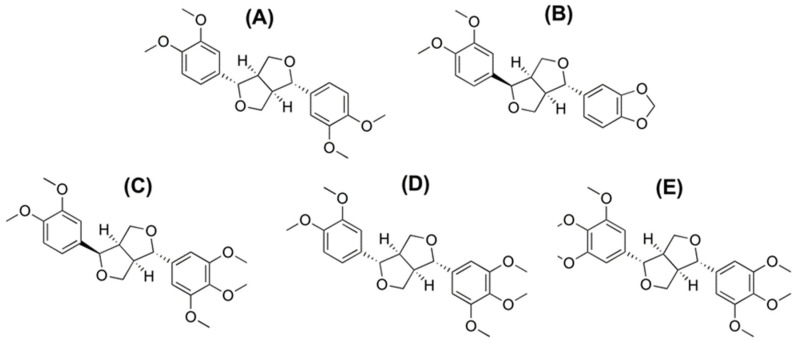
Chemical structures of (**A**) eudesmin, (**B**) fargesin, (**C**) epimagnolin A, (**D**) magnolin, and (**E**) yangambin.

**Figure 2 pharmaceutics-13-00187-f002:**
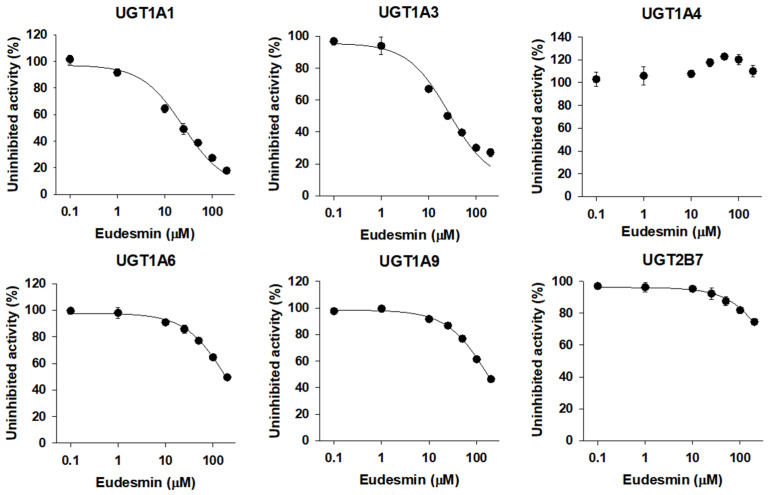
Inhibitory effects of eudesmin (0.1–200 µM) on the UGT1A1-, UGT1A3-, UGT1A4-, UGT1A6-, UGT1A9-, and UGT2B7-mediated glucuronidation of 0.5 µM SN-38, 2 µM chenodeoxycholic acid, 0.5 µM trifluoperazine, 1 µM *N*-acetylserotonin, 0.2 µM mycophenolic acid, and 1 µM naloxone, a probe substrate for UGT1A1, UGT1A3, UGT1A4, UGT1A6, UGT1A9, and UGT2B7, respectively, in human liver microsomes. The data are expressed as means ± standard deviations (*n* = 3).

**Figure 3 pharmaceutics-13-00187-f003:**
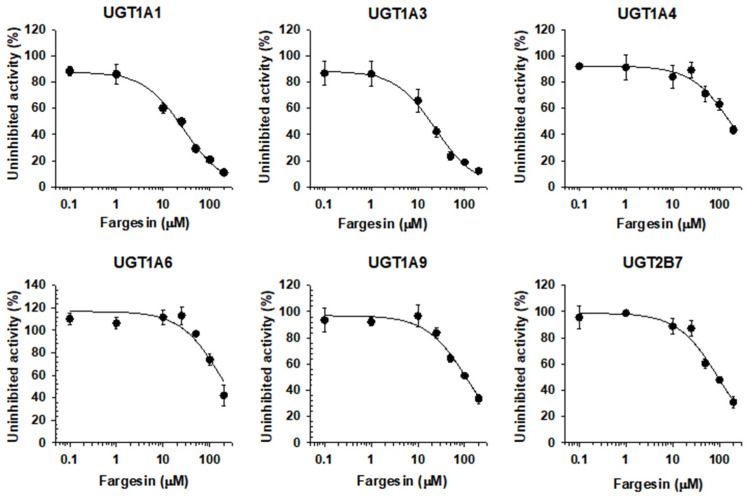
Inhibitory effects of fargesin (0.1–200 µM) on UGT1A1-, UGT1A3-, UGT1A4-, UGT1A6-, UGT1A9-, and UGT2B7-mediated glucuronidation of 0.5 μM SN-38, 2 µM chenodeoxycholic acid, 0.5 µM trifluoperazine, 1 µM *N*-acetylserotonin, 0.2 µM mycophenolic acid, and 1 µM naloxone, a probe substrate for UGT1A1, UGT1A3, UGT1A4, UGT1A6, UGT1A9, and UGT2B7, respectively, in human liver microsomes. The data are expressed as means ± standard deviations (*n* = 3).

**Figure 4 pharmaceutics-13-00187-f004:**
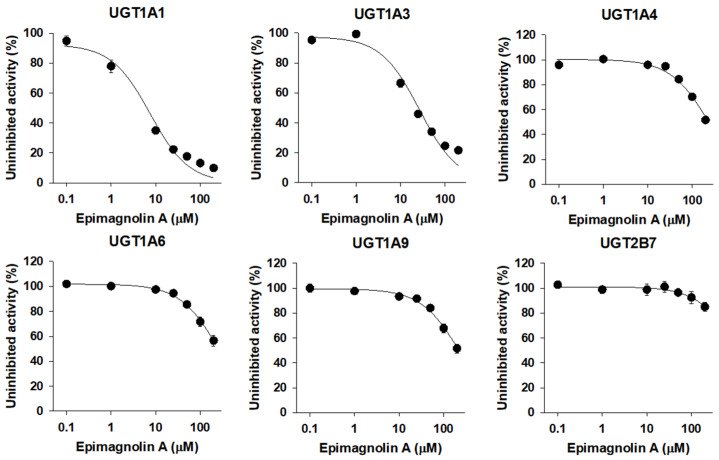
Inhibitory effects of epimagnolin A (0.1–200 µM) on UGT1A1-, UGT1A3-, UGT1A4-, UGT1A6-, UGT1A9-, and UGT2B7-mediated glucuronidation of 0.5 μM SN-38, 2 µM chenodeoxycholic acid, 0.5 µM trifluoperazine, 1 µM *N*-acetylserotonin, 0.2 µM mycophenolic acid, and 1 µM naloxone, a probe substrate for UGT1A1, UGT1A3, UGT1A4, UGT1A6, UGT1A9, and UGT2B7, respectively, in human liver microsomes. The data are expressed as means ± standard deviations (*n* = 3).

**Figure 5 pharmaceutics-13-00187-f005:**
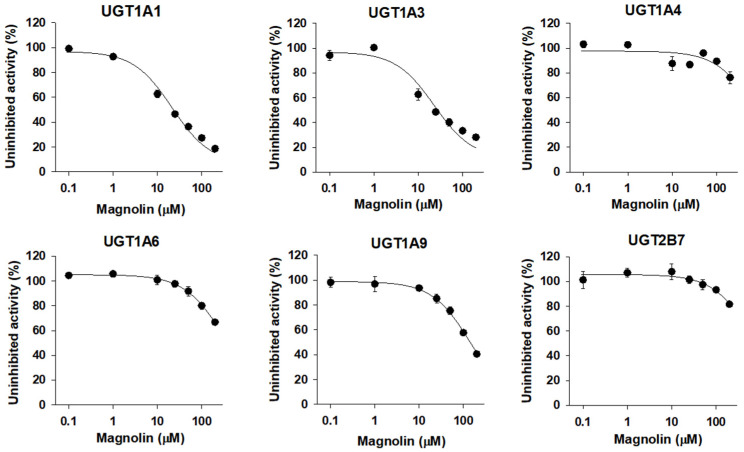
Inhibitory effects of magnolin (0.1–200 µM) on UGT1A1-, UGT1A3-, UGT1A4-, UGT1A6-, UGT1A9-, and UGT2B7-mediated glucuronidation of 0.5 μM SN-38, 2 µM chenodeoxycholic acid, 0.5 µM trifluoperazine, 1 µM *N*-acetylserotonin, 0.2 µM mycophenolic acid, and 1 µM naloxone, a probe substrate for UGT1A1, UGT1A3, UGT1A4, UGT1A6, UGT1A9, and UGT2B7, respectively, in human liver microsomes. The data are expressed as means ± standard deviations (*n* = 3).

**Figure 6 pharmaceutics-13-00187-f006:**
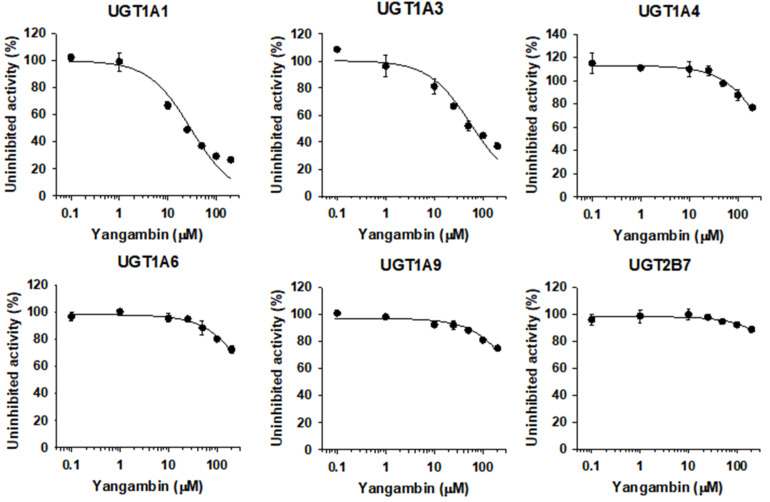
Inhibitory effects of yangambin (0.1–200 µM) on UGT1A1-, UGT1A3-, UGT1A4-, UGT1A6-, UGT1A9-, and UGT2B7-mediated glucuronidation of 0.5 μM SN-38, 2 µM chenodeoxycholic acid, 0.5 µM trifluoperazine, 1 µM *N*-acetylserotonin, 0.2 µM mycophenolic acid, and 1 µM naloxone, a probe substrate for UGT1A1, UGT1A3, UGT1A4, UGT1A6, UGT1A9, and UGT2B7, respectively, in human liver microsomes. The data are expressed as means ± standard deviations (*n* = 3).

**Figure 7 pharmaceutics-13-00187-f007:**
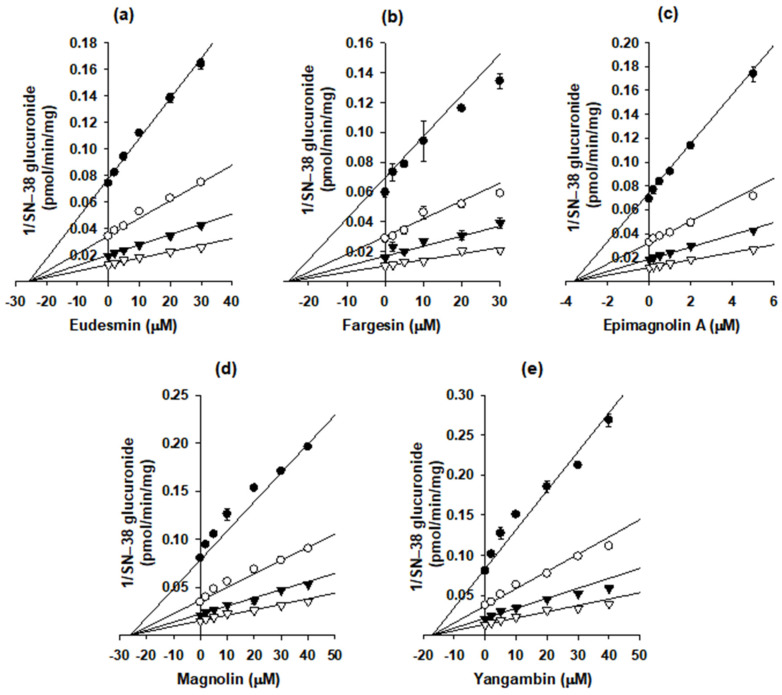
Dixon plots for the inhibitory effects of (**a**) eudesmin, (**b**) fargesin, (**c**) epimagnolin A, (**d**) magnolin, and (**e**) yangambin on UGT1A1-catalyzed SN-38 glucuronidation in pooled human liver microsomes. Each symbol represents the concentration of SN-38: ●, 0.2 μM; ○, 0.5 μM; ▼, 1 μM; ▽, 2 μM. The data are means ± SDs (*n* = 3).

**Figure 8 pharmaceutics-13-00187-f008:**
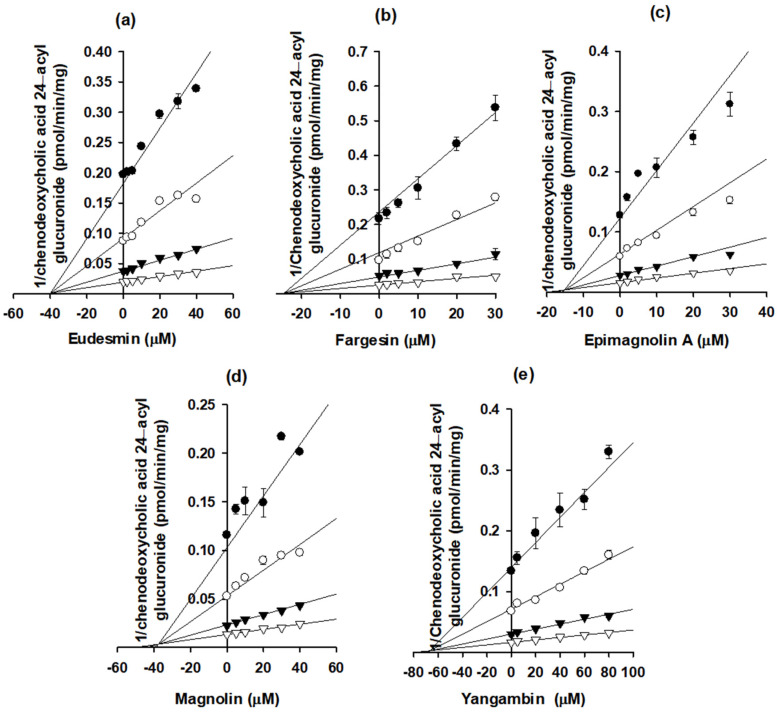
Dixon plots for the inhibitory effects of (**a**) eudesmin, (**b**) fargesin, (**c**) epimagnolin A, (**d**) magnolin, and (**e**) yangambin on UGT1A3-catalyzed chenodeoxycholic acid 24-acyl glucuronidation in pooled human liver microsomes. Each symbol represents the concentration of chenodeoxycholic acid: ●, 1 μM; ○, 2 μM; ▼, 5 μM; ▽, 10 μM. The data are means ± SDs (*n* = 3).

**Table 1 pharmaceutics-13-00187-t001:** Inhibitory potentials of eudesmin, fargesin, epimagnolin A, magnolin, and yangambin on six major UGT enzyme activities in ultrapooled human liver microsomes.

UGT	Enzyme Activity	IC_50_ (µM)				
		Eudesmin	Fargesin	Epimagnolin A	Magnolin	Yangambin
1A1	SN-38 glucuronidation	24.3	24.7	7.5	21.3	29.7
1A3	chenodeoxycholic acid 24-acyl-glucuronidation	26.6	21.5	26.6	22.9	56.5
1A4	trifluoperazine *N*-glucuronidation	>200	182.7	>200	>200	>200
1A6	*N*-acetylserotonin glucuronidation	195.6	193.9	>200	>200	>200
1A9	mycophenolic acid glucuronidation	173.2	110.9	>200	145.7	>200
2B7	naloxone 3-β-D-glucuronidation	>200	94.7	>200	>200	>200

**Table 2 pharmaceutics-13-00187-t002:** Kinetic parameters for the inhibition of eudesmin, fargesin, epimagnolin A, magnolin, and yangambin on UGT1A1 and UGT1A3 enzyme activities in pooled human liver microsomes.

	*K_i_* (μM, Inhibition Mode)	
	SN-38 Glucuronidation (UGT1A1)	Chenodeoxycholic Acid 24-acyl-Glucuronidation (UGT1A3)
Eudesmin	25.7 (noncompetitive)	39.8 (competitive)
Fargesin	25.3 (noncompetitive)	24.5 (competitive)
Epimagnolin A	3.6 (noncompetitive)	15.1 (competitive)
Magnolin	26.0 (noncompetitive)	37.6 (competitive)
Yangambin	17.1 (noncompetitive)	66.8 (competitive)

## Data Availability

The data presented in this study are available upon request.

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
