# Peer review of "Tetrahydrofurofuranoid Lignans, Eudesmin, Fargesin, Epimagnolin A, Magnolin, and Yangambin Inhibit UDP-Glucuronosyltransferase 1A1 and 1A3 Activities in Human Liver Microsomes"

_pharmaceutics, 2021, doi:10.3390/pharmaceutics13020187_

Round 1
Reviewer 1 Report
Manuscript Number: Pharmaceutics-1046961
Title: Tetrahydrofurofuranoid Lignans, Eudesmin, Fargesin, Epimagnolin A, Magnolin, and Yangambin Inhibit UDP-glucuronosyltransferase 1A1 and 1A3 Activities in Human Liver Microsomes
Comments to Authors
Recommendation: Major revision
The article " Tetrahydrofurofuranoid Lignans, Eudesmin, Fargesin, Epimagnolin A, Magnolin, and Yangambin Inhibit UDP-glucuronosyltransferase 1A1 and 1A3 Activities in Human Liver Microsomes” aimed to investigate the in vitro inhibitory potentials and inhibition kinetics of bioactive five tetrahydrofurofuranoid lignans on UGT1A1, UGT1A3, UGT1A4, UGT1A6, UGT1A9, and UGT2B7 activities using human liver microsomes.
Although the presented work is well written and well designed, the used methodology appropriate, prior to publishing, several adjustments and corrections should be made:
Taking into account that authors in Introduction part mentioned three plants as potential source of the investigated lignans (“…Flos Magnoliae, Aristolochia elegans rhizomes, and Zanthoxylum armatum DC….”), authors are asked to be specific ‘- to connect the biological source and its chemical profile – in the presented form, it might be assumed that all three mentioned biological sources are rich in all investigated lignans.
When the plant has been mentioned for the first time, the full Latin name should be provided: Latin name, author abbreviation, and Family, while full name should be in Italic. Afterwards, authors might switch to shorten form, or common name.
Page 1, line 36, please, give the biological name of Magnoliae flos
Page 1, lines 36-37, please give the full Latin name of mentioned Aristolochia sp. If authors would like to point the part of the plants investigated, then the Latin name should be in genitive, followed by the the part of the plant used. Please, correct Aristolochia elegans rhizomes
Page 1, line 37. please, give the full Latin name
Please, give the full proper Latin name for Page 2, lines 64,56; lines 70, 71
As Authors used the data regarding the patent extract NDC-052, the chemical composition, qualitative and quantitative, regarding the lignans of the interest should be given.
Authors started from the fact that all investigated lignans will be present in plasma. What is the connection between the investigated concentration of the five lignans and their amount when applied within the Magnoliae flos extract, named as NDC-052?
Please, connect the used investigated lignans concentrations (0.1-200µM) in the experiment with the data cited from literature – that “…the single oral administrations of NDC-052 (22 mg/kg) to rats resulted in the maximum plasma concentration of epimagnolin A, fargesin, and magnolin of 3.07 ± 0.72 µM, 0.38 ± 0.18 µM, and 5.95 ± 1.12 µM, respectively…” while the eudesmin and yangambin were not determined, not even in the repeated experiments of other investigators with doses of 50 mg/kg/dayfor 8 weeks in guinea pig experiments and in adults asthmatic patients (600mg/day, 8 weeks).
Please, specify “….the reactive intermediate…”, page 11, line 291
Please, specify “…other components…”, page 11, line 302
In Discussion section, authors should provide the scientific explanation why the mentioned lignans were chosen.
Author Response
1. Taking into account that authors in Introduction part mentioned three plants as potential source of the investigated lignans (“…Flos Magnoliae, Aristolochia elegans rhizomes, and Zanthoxylum armatumDC….”), authors are asked to be specific ‘- to connect the biological source and its chemical profile – in the presented form, it might be assumed that all three mentioned biological sources are rich in all investigated lignans.When the plant has been mentioned for the first time, the full Latin name should be provided: Latin name, author abbreviation, and Family, while full name should be in Italic. Afterwards, authors might switch to shorten form, or common name.
Page 1, line 36, please, give the biological name of Magnoliae flos
Page 1, lines 36-37, please give the full Latin name of mentioned Aristolochia sp. If authors would like to point the part of the plants investigated, then the Latin name should be in genitive, followed by the the part of the plant used. Please, correct Aristolochia elegans rhizomes
Page 1, line 37. please, give the full Latin name
Answer> Thanks for the reviewer’s valuable comment. We totally agree with the reviewer’s comments. Since Aristolochia elegans rhizomes is reported to contain fagesin but magnolin, epimagnolin A, and yangambin do not exist in Aristolochia elegans rhizomes. Therefore, we deleted Aristolochia elegans rhizomes and corrected the other plant name according to the reviewer’s comment as follows:
(Page 1, line 37-38) Eudesmin, fargesin, epimagnolin A, magnolin, and yangambin (Figure 1) are the pharmacologically active tetrahydrofurofuranoid lignans found in Magnolia denudata Desrousseaux, Magnolia fargesii Cheng (family: Magnoliaceae), and Zanthoxylum armatum DC (family: Rutaceae) [1-5].
2. Please, give the full proper Latin name for Page 2, lines 64,56; lines 70, 71
Answer> We corrected the other plant name according to the reviewer’s comment as follows:
(Page 2, line 50-54) Eudesmin, fargesin, epimagnolin A, magnolin, and yangambin (Figure 1) are the pharmacologically active tetrahydrofurofuranoid lignans found in Magnolia denudata Desrousseaux, Magnolia fargesii Cheng (family: Magnoliaceae), and Zanthoxylum armatum DC (family: Rutaceae) [1-5].
(Page 2, line 70-71) The herb-drug interactions among the herbal drugs (e.g., Hypericum perforatum, Ginkgo biloba, Camellia sinensis, Glycyrrhiza glabra, Allium sativum, Coptis chinensis, and Silybum marianum)
(Page 2, line 76) Aschantin, a bioactive tetrahydrofurofuranoid lignan found in Magnolia biondii and Hernandia nymphaeifolia,......
3. As Authors used the data regarding the patent extract NDC-052, the chemical composition, qualitative and quantitative, regarding the lignans of the interest should be given. Authors started from the fact that all investigated lignans will be present in plasma. What is the connection between the investigated concentration of the five lignans and their amount when applied within the Magnoliae flos extract, named as NDC-052?
Answer> We added the chemical composition and quantitative analysis results in the revised manuscript.
(Page 2, line 50-52) The ethanol extract of the dried flower buds of Magnolia fargesii Cheng (encoded as NDC-052 contained eudesmin, fargesin, epimagnolin A, magnolin, and yangambin as 4.1, 3.4, 11.9, 21.5, and 9.1%, respectively, as determined by the LC-APCI-MS/MS method [18]) ~~
4. Please, connect the used investigated lignans concentrations (0.1-200 µM) in the experiment with the data cited from literature – that “…the single oral administrations of NDC-052 (22 mg/kg) to rats resulted in the maximum plasma concentration of epimagnolin A, fargesin, and magnolin of 3.07 ± 0.72 µM, 0.38 ± 0.18 µM, and 5.95 ± 1.12 µM, respectively…” while the eudesmin and yangambin were not determined, not even in the repeated experiments of other investigators with doses of 50 mg/kg/day for 8 weeks in guinea pig experiments and in adults asthmatic patients (600mg/day, 8 weeks).
Answer> Thanks for the reviewer’s valuable comments. We added the plasma concentrations of eudesmin and yangambin in addition to those of epimagnolin A, fargesin, and magnolin in the revised manuscript. We also added the expected human gastrointestinal concentrations of these five lignans by considering the content in the NDC-052 and oral dilution volume in human (500–2200 mL). Thus, the expected gastric concentrations of five lignans were ranged from 25–618 µM. In addition, we discussed the concentrations that we used for the inhibition study regarding the plasma and gastric concentrations of five lignans.
(Page 11, line 274) The concentration range of lignans tested in this study could cover the plasma concentration of lignans in rats after an oral administration of NDC-052 (22 mg/kg) and the expected gastric concentrations of five lignans. For example, the Cmax values of eudesmin, fargesin, epimagnolin A, magnolin, and yangambin were 2.2 ± 0.41 μM, 0.38 ± 0.18 μM, 3.07 ± 0.72 μM, 5.95 ± 1.12 μM, and 0.52 ± 0.30 μM, respectively, after single oral administration of NDC-052 (22 mg/kg) [18]. Herbal extracts are diluted by gastrointestinal fluid after oral administration by a factor of oral dilution volume ((500–2200 mL in humans) [32, 33]. Therefore, the expected gastric concentrations of eudesmin, fargesin, epimagnolin A, magnolin, and yangambin could be calculated as 29-127 μM, 25-110 μM, 78-342 μM, 140-618 μM, and 55-244 μM by considering the content of five lignans (eudesmin, fargesin, epimagnolin A, magnolin, and yangambin as 4.1, 3.4, 11.9, 21.5, and 9.1%, respectively) [18] and oral dose of NDC-052 in adult asthmatic patients (600 mg/day for 8 weeks) [19].
5. Please, specify “….the reactive intermediate…”, page 11, line 291
Answer> We revised the expression as follows to specify the meaning of the reactive intermediate metabolites:
(Page 11, line 310-314) Consequently, it indicated that CYP2C8, CYP2C9, CYP2C19, and CYP3A4 activities might be inhibited by the presence of fargesin that may cause reversible inhibition on CYP2C8 and the formation of intermediated metabolites of fargesin which may cause irreversible inhibition of CYP2C9, CYP2C19, and CYP3A4.
6. Please, specify “…other components…”, page 11, line 302
Answer> We deleted “other components” to avoid confusion:
(Page 12, line 324) Taken togenther, fargesin may be involved in the two-fold increase of magnolin AUC by inhibiting the CYP2C8,…
7. In Discussion section, authors should provide the scientific explanation why the mentioned lignans were chosen.
Answer> We added the reason for the selection of five lignans in the discussion section:
(Page 10, line 265-271) Eudesmin, fargesin, epimagnolin A, magnolin, and yangambin were selected as major tetrahydrofurofuranoid lignans that exist in the ethanol extract of flower bud of Magnolia fargesii (NDC-052) and also are present in the rat plasma samples following oral administration [18]. Therefore, the inhibitory potentials (IC50 values) of eudesmin, fargesin, epimagnolin A, magnolin, and yangambin on UGT1A1, UGT1A3, UGT1A4, UGT1A6, UGT1A9, and UGT2B7 enzyme activities were evaluated in pooled human liver microsomes in the concentration ranges of 0.1–200 μM of individual tetrahydrofurofuranoid lignan (Figures 2-6; Table 1).

Reviewer 2 Report
The authors of this manuscript evaluated the inhibitory effects of multiple bioactive phytochemicals on the enzymatic activities of UGT isoforms. Proper justification was provided for the study. The evaluation of herb-drug interaction potential usually start with the assessment of the herbal product on CYP activities. UGTs are important, but not as important as CYPS. The authors however, showed that studies with CYPs have been done on these phytochemicals. Thus, this study is complementary and provide readers with the missing information (effect on UGTS) relevant for the assessment of herb-drug interaction potential.
The methodologies employed in this study are scientifically sound. The analysis and discussions are reasonable. The conclusions are fair and reflective of the results.
Author Response
The authors of this manuscript evaluated the inhibitory effects of multiple bioactive phytochemicals on the enzymatic activities of UGT isoforms. Proper justification was provided for the study. The evaluation of herb-drug interaction potential usually start with the assessment of the herbal product on CYP activities. UGTs are important, but not as important as CYPS. The authors however, showed that studies with CYPs have been done on these phytochemicals. Thus, this study is complementary and provide readers with the missing information (effect on UGTS) relevant for the assessment of herb-drug interaction potential.
The methodologies employed in this study are scientifically sound. The analysis and discussions are reasonable. The conclusions are fair and reflective of the results.
Answer> We appreciated the reviewer’s positive review.
Reviewer 3 Report
The paper is not suitable for Pharmaceutics because it does not match with the scope and aims of this journal. In addition, the study is too preliminary and routinely because it reports on in vitro tests of commercial natural products. The manuscript could be more appropiate for other developing MDPI journals having in the scope these screening tests, i.e. Scientia Pharmaceutica
Author Response
The paper is not suitable for Pharmaceutics because it does not match with the scope and aims of this journal. In addition, the study is too preliminary and routinely because it reports on in vitro tests of commercial natural products. The manuscript could be more appropiate for other developing MDPI journals having in the scope these screening tests, i.e. Scientia Pharmaceutica
Answer> We could not agree with the reviewer’s point.
Eudesmin, fargesin, epimagnolin A, magnolin, and yangambin were selected as major tetrahydrofurofuranoid lignans that are exist in the ethanol extract of flower bud of Magnolia fargesii (encoded as NDC-052 at a content of 4.1, 3.4, 11.9, 21.5, and 9.1%, respectively, as determined by the LC-APCI-MS/MS method [18]). These five lignans are present in the rat plasma samples following oral administration of NDC-052 [18]. NDC-052 has been developed in Korea as an effective alternative or complement to standard asthma therapy based on their biological activities. Consequently, taking NDC-052 (600 mg/day for 8 weeks per oral) in adult asthmatic patients was safe and tolerated [19]. The add-on therapy of NDC-052 (600 mg/day for 8 weeks per oral) with inhaled corticosteroids in asthmatic patients had a beneficial effect on asthma control [19]. The pharmacokinetics of major tetrahydrofurofuranoid lignans in rats were also investigated in addition to their biological activities. From these studies, we found the possibility of drug interactions among the constituents of herbal drugs. Subsequently, we investigated the inhibition potencies of eudesmin, fargesin, epimagnolin A, magnolin, and yangambin on six major human uridine 5′-diphospho-glucuronosyltransferase (UGT) activities in this study in addition to our previous studies on CYP enzymes [30].
We submitted our manuscript to the special issue entitled "Transport and Metabolism of Small-Molecule Drugs" of Pharmaceutics. We believe that our manuscript fall into the scope of this special issue and this study would provide the molecular mechanism for understanding the drug interactions in relation to UGTs among the structurally related five tetrahydrofurofuranoid lignans and the concomitantly administered victim drugs although clinical herb-drug interaction studies await further investigation.
Also, the manuscript submitted to Pharmaceutics has been pre-checked by an academic editor prior to the peer review process, based on our experiences of academic editors in Pharmaceutics.
[18] Jeong, J.H.; Kim, D.K.; Ji, H.Y.; Oh, S.R.; Lee, H.K.; Lee, H.S. Liquid chromatography-atmospheric pressure chemical ionization tandem mass spectrometry for the simultaneous determination of dimethoxyaschantin, dimethylliroresinol, dimethylpinoresinol, epimagnolin A, fargesin and magnolin in rat plasma. Biomed. Chromatogr. 2011, 25, 879-889.
[19] Park, C.S.; Kim, T.B.; Lee, J.Y.; Park, J.Y.; Lee, Y.C.; Jeong, S.S.; Lee, Y.D.; Cho, Y.S.; Moon, H.B. Effects of add-on therapy with NDC-052, an extract from Magnoliae flos, in adult asthmatic patients receiving inhaled corticosteroids. Kor. J. Intern. Med. 2012, 27, 84-90.
[30] Kim, J.H.; Kwon, S.S.; Jeong, H.U.; Lee, H.S. Inhibitory Effects of Dimethyllirioresinol, Epimagnolin A, Eudesmin, Fargesin, and Magnolin on Cytochrome P450 Enzyme Activities in Human Liver Microsomes. Int. J. Mol. Sci. 2017, 18, 952.
Round 2
Reviewer 1 Report
Manuscript Number: pharmaceutics-1046961
Comments to Authors
Recommendation: Major revision
The authors “Tetrahydrofurofuranoid Lignans, Eudesmin, Fargesin, Epimagnolin A, Magnolin, and Yangambin Inhibit UDP-glucuronosyltransferase 1A1 and 1A3 Activities in Human Liver Microsomes” succeeded tomake some changes suggested by the reviewers.
Although improved, the manuscript still could not be considered for publication in the Journal, especially as the IF of Pharmaceutics indicated the journal of great impact and importance in the field of pharmaceutical formulation, process development, drug delivery, pharmacokinetics, biopharmaceutics, pharmacogenetics
Please, put in Italic bacterial strains
Authors are asked to give the full Latin name of all plants mentioned in the manuscript
After giving the full Latin name, the short form of the name of the plants and microbial strains should be used
Please, give the information of the used concentrations of the investigated compounds
Please, explain why the activity of the mixture of the investigated compounds was not evaluated (especially the mixture containing the investigated specific compounds in the quantity present in the mentioned composition, NDC-052 )
In the presented form, the authors showed that each of the investigated compounds exhibited certain potential on enzymatic activity, but in the mentioned ethanol extract encoded as NDC-052 all of the tested compounds were present jointly.
It is necessary to determine the potential of the mixture on the investigated enzymatic activity
Author Response
Thanks for the reviewer's valuable comments. The reviewer’s comments were reflected accordingly during the revision.
The point-by-point responses to the reviewer’s comments (2nd round)
The authors “Tetrahydrofurofuranoid Lignans, Eudesmin, Fargesin, Epimagnolin A, Magnolin, and Yangambin Inhibit UDP-glucuronosyltransferase 1A1 and 1A3 Activities in Human Liver Microsomes” succeeded to make some changes suggested by the reviewers.
Although improved, the manuscript still could not be considered for publication in the Journal, especially as the IF of Pharmaceutics indicated the journal of great impact and importance in the field of pharmaceutical formulation, process development, drug delivery, pharmacokinetics, biopharmaceutics, pharmacogenetics.
1. Please, put in Italic bacterial strains
Answer> In this study, we investigated the inhibitory effect of eudesmin, fargesin, epimagnolin A, magnolin, or yangambin on 6 UGT isozyme activities in human liver microsomes. The five lignans used in this study were purchased from PhytoLab GmbH & Co. (Vestenbergsgreuth, Germany), Selleck Chemicals LLC. (Houston, TX, USA), and ChemFaces Biochemical Co. Ltd. (Wuhan, China) and NDC-052, an ethanol extract of the dried flower buds of M. fargesii, was also provided from Korea Plant Extract Bank, Korea Research Institute of Biology and Biotechnology (Daejeon, Korea). Therefore, we did not use bacterial strains for the preparation of investigated compounds.
2. Authors are asked to give the full Latin name of all plants mentioned in the manuscript. After giving the full Latin name, the short form of the name of the plants and microbial strains should be used
Answer> As the reviewer commented, we used the full latin names of all plant at their first appearance and, then used the short form of the plant name after their second appearance. As we previously mentioned, we did not use microbial strains in this study.
(Page 2, line 37) … tetrahydrofurofuranoid lignans found in Magnolia denudata Desrousseaux, M. fargesii Cheng (family: Magnoliaceae),…
(Page 2, line 51) The ethanol extract of the dried flower buds of M. fargesii …..
(Page 2, line 77) Aschantin, a bioactive tetrahydrofurofuranoid lignan found in M. biondii and ….
3. Please, give the information of the used concentrations of the investigated compounds
Answer> As the reviewer commented, we added the concentrations of the five lignans in the method section and we also added the concentration range of these five lignans in the results and figure legends.
(Page 3, line 120) various concentrations of eudesmin, fargesin, epimagnolin A, magnolin, or yangambin in acetonitrile (final concentrations of 0.1, 1, 10, 20, 50, 100, 200 µM each, acetonitrile less than 1% (v/v)),
(lines 176, 184, 190, 197, 205, 211, 217, 225, 231, and 237) … Inhibitory effects of eudesmin, fargesin, epimagnolin A, magnolin, or yangambin (0.1–200 µM) on UGT1A1-, UGT1A3-…
4. Please, explain why the activity of the mixture of the investigated compounds was not evaluated (especially the mixture containing the investigated specific compounds in the quantity present in the mentioned composition, NDC-052). In the presented form, the authors showed that each of the investigated compounds exhibited certain potential on enzymatic activity, but in the mentioned ethanol extract encoded as NDC-052 all of the tested compounds were present jointly. It is necessary to determine the potential of the mixture on the investigated enzymatic activity
Answer> The purpose of this study was to investigate the inhibition potencies of eudesmin, fargesin, epimagnolin A, magnolin, and yangambin on six major human UGT activities in human liver microsomes and to investigate the mode of inhibition of the five tetrahydrofurofuranoid lignans on the affected UGT isozymes for the elucidation of drug-drug interaction potential of the above lignans with UGTs because in vivo drug-drug interaction potential could be predicted with kinetic parameters and inhibition mode of the inhibitors as well as the plasma concentrations of the inhibitors [references 1, 2, 3]. From the present in vitro inhibition study, we identified that eudesmin, fargesin, epimagnolin A, magnolin, and yangambin inhibited UGT1A1 and UGT1A3 activities in a noncompetitive and competitive mode, respectively, and we calculated Ki values of these five lignans. To evaluate the drug-drug interaction potential of these five lignans on UGT1A1 and UGT1A3, we discussed the composition and the plasma concentrations of NDC-052 extract as an example of pharmacologically active herbal supplements that contained these five lignans. We did not perform the in vitro inhibitory effect of NDC-052 on UGT activities at first time because the composition of eudesmin, fargesin, epimagnolin A, magnolin, and yangambin accounted 50% of the NDC-052 extract and unveiled components accounted for another 50% of NDC-052 and the unveiled components may serve as inhibitors of UGT enzymes [reference 42].
However, we also agree with the reviewer’s point of view. Thus, we investigated the in vitro inhibitory effect of NDC-052 on the UGT1A1- and UGT1A3-mediated glucuronidation in human liver microsomes during the revision according to the reviewer’s comments.
The methods, results, and discussion were amended accordingly in the revised manuscript as follows.
(Page 4, line 138) The inhibitory effect of NDC-052 on UGT1A1 and UGT1A3 was also measured by incubating the incubation mixtures (100 µL) including 50 mM Tris buffer (pH 7.4), human liver microsomes (0.1 mg/mL), SN-38 (0.5 µM) for UGT1A1 or chenodeoxycholic acid (2 µM) for UGT1A3, 5 mM UDPGA, 25 µg/mL alamethicin, 10 mM MgCl2, and various concentrations of ND-052 (0.1-100 µg/mL) for 30 min.
(Page 8, line 252) Next, we measured the IC50 values of NDC-052 that contained eudesmin, fargesin, epimagnolin A, magnolin, and yangambin at a content of 4.1, 3.4, 11.9, 21.5, and 9.1%, respectively [18] on UGT1A1- and UGT1A3-mediated glucuronidation. The UGT1A1 and UGT1A3 activities were inhibited by the presence of NDC-052 (0.1–100 µg/mL) in a concentration dependent manner and the IC50 values of NDC-052 for UGT1A1 and UGT1A3 were calculated as 38.1 and 65.0 µg/mL, respectively.
(Page 11, line 305) As expected, NDC-052 inhibited UGT1A1- and UGT1A3-mediated glucuronidation in a concentration dependent manner with IC50 values of 38.1 and 65.0 µg/mL, respectively. When the IC50 values were calculated as the concentration of individual lignans by using the content of five lignans in NDC-052, 38.1 and 65.0 µg/mL of NDC-052 contained 4.0 and 6.9 µM of eudesmin, 3.5 and 6.0 µM of fargesin, 10.9 and 18.6 µM of epimagnolin A, 19.7 and 33.6 µM of magnolin, and 7.8 and 13.2 µM of yangambin, respectively. These concentrations were lower than the IC50 values of individual lignans in Table 1, which suggested that the inhibitory effect of NDC-052 on the metabolic activities of UGT1A1 and UGT1A3 may be caused by the coexistence of lignans rather than the inhibition by single lignan. However, we also should note that the total content of five lignans accounts for 50% of NDC-052 and unveiled components such as terpenoids, alkaloids, and flavonoids may serve as inhibitors of UGT enzymes [42].
References
[1] Adiwidjaja, J.; Boddy, A.V.; McLachlan, A.J. Physiologically based pharmacokinetic modelling of hyperforin to predict drug interactions with St John's wort. Clin. Pharmacokinet. 2019, 58, 911–926.
[2] Brantley, S.; Graf, T.N.; Oberlies, N.H.; Paine, M.F. Physiologically-based pharmacokinetic interaction model framework for quantitative prediction of herb-drug interactions. Drug Metab. Rev. 2012, 44, 99–100.
[3] Brantley, S.J.; Argikar, A.A.; Lin, Y.S.; Nagar, S.; Paine, M.F. Herb–drug interactions: Challenges and opportunities for improved predictions. Drug Metab. Dispos. 2014, 42, 301-317.
[42] Shen, Y.; Li, C.G.; Zhou, S.F.; Pang, E.C.; Story, D.F.; Xue, C.C. Chemistry and bioactivity of Flos Magnoliae, a Chinese herb for rhinitis and sinusitis. Curr. Med. Chem. 2008, 15, 1616-1627.
Round 3
Reviewer 1 Report
Comment on Manuscript No: pharmaceutics-1046961
Authors improved the revised version of the manuscript.